# Bacterial and Microeukaryotic Community Compositions and Their Assembly Processes in Lakes on the Eastern Qinghai-Tibet Plateau

**DOI:** 10.3390/microorganisms12010032

**Published:** 2023-12-23

**Authors:** Dandan Wang, Yuefei Huang, Haichao Jia, Haijiao Yang

**Affiliations:** 1School of Civil Engineering and Water Resources, Qinghai University, Xining 810016, China; 2023990002@qhu.edu.cn (D.W.); jhc5914930722023@163.com (H.J.); yyeezzii@163.com (H.Y.); 2Key Laboratory of Ecological Protection and High Quality Development in the Upper Yellow River, Qinghai University, Xining 810016, China; 3Key Laboratory of Water Ecological Remediation and Protection at Headwater Regions of Big Rivers, Qinghai University, Xining 810016, China; 4State Key Laboratory of Plateau Ecology and Agriculture, Qinghai University, Xining 810016, China

**Keywords:** Qinghai-Tibet Plateau, lake ecosystem, community assembly processes, co-occurrence network

## Abstract

Bacterial and microeukaryotic community compositions and their assembly processes have remained challenging and remained unclear in lake ecosystems on the Qinghai-Tibet Plateau (QTP). We revealed the diversity and community compositions, driving factors, ecological assembly processes, and co-occurrence networks of bacterial and microeukaryotic communities in water bodies of the eight lake ecosystems across the Eastern QTP. The results demonstrated that the predominant bacteria in most samples were Proteobacteria, with an average relative abundance of 41.78%, whereas the most abundant of microeukaryotes differed among the sample sites. The redundancy analysis revealed that latitude and pH were the most important driving factors in shaping the bacterial and microeukaryotic community compositions. Homogeneous selection (56.40%) was the dominant process in assembling the bacterial communities, whereas dispersal limitation (67.24%) was the major process in governing the microeukaryotic communities. Furthermore, dissolved organic carbon and salinity were the major factors mediating the balance of deterministic and stochastic assembly processes in the bacterial and microeukaryotic communities. Both the bacterial and microeukaryotic community co-occurrence networks exhibited topological features of modularity and non-random topological features. The results offer insights into the mechanisms underpinning bacterial and microeukaryotic diversities and communities in the lake ecosystems on the QTP.

## 1. Introduction

The Qinghai-Tibet Plateau (QTP), with a large number of naturally developed lakes, is often referred to as “the third pole” or “the roof of the world” due to its extremely harsh environmental conditions, including a high altitude, low temperatures, strong radiation, and oligotrophic environment. Microbiota living in the lakes on the QTP are considered good sentinels of climate change and anthropogenic activities and play pivotal roles in the compositions, functions, and biogeochemical cycles of the high-altitude lake ecosystems [1,2,3,4]. Thus, elucidating microbial assembly processes is essential for understanding the underlying mechanisms of microbial community compositions and their ecological functions in the high-altitude lakes. However, they have not been sufficiently explored, and there is a longstanding challenge to understanding the mechanisms governing microbial diversity, community composition, driving factors, assembly processes, and co-occurrence networks in the lake ecosystems on the QTP. 

To date, numerous studies have used co-occurrence networks to analyze species–species interactions of microbiota in the lake ecosystems [2,5]. A previous study suggested that the complexity of the bacterial community network showed a positive relationship with the salinity of the Tibetan Plateau lakes [5]. Subsequently, another work reported that the prokaryotic community in the waters and sediments of the lakes across the Qaidam Basin had different co-occurrence patterns [6]. Overall, previous studies have mostly focused on bacterial co-occurrence patterns, and fewer studies have considered the bacterial and microeukaryotic communities simultaneously and how microbial co-occurrence network structures (i.e., average degree) are affected by in situ environmental factors in the high-altitude lakes. 

Characterizing the mechanisms of microbial community assembly that support microbial diversities, community compositions, functions, distributions, and successions is an important theme in microbial ecology [7]. It has been demonstrated that deterministic processes (referred to as abiotic factors and interspecies correlations) and stochastic processes (considering birth, death, and immigration rates) are two major types of ecological processes explaining microbial community assembly dynamics [8,9,10,11], and many studies have quantified the microbial community assembly processes of various habitats, such as reservoirs, springs, lakes, and rivers [11,12,13,14,15,16]. As reported by several previous researchers, deterministic processes (i.e., homogeneous selection) are the dominant processes in assembling microbial community compositions in aquatic habitats [12,17,18]. More recently, dispersal limitation has been proven to be the dominant process in shaping bacterial and microeukaryotic metacommunities in rock pools [19]. Further, previous studies have revealed that free-living and particle-attached bacterial communities have different assembly processes in different seasons in the lake ecosystems on the QTP [18,20]. Likewise, the environmental filtering and dispersal limitation processes to determine the bacterial and microeukaryotic communities in lakes in the southwestern parts of the QTP have also been carried out [21]. However, fewer studies, to date, have focused on the influence of the shifting environmental factors on the balance of the deterministic and stochastic processes of microbial community formation in lake ecosystems on the QTP. Hence, the specific objectives of this study are as follows: (i) to describe the diversity, community compositions, and the driving factors of the bacterial and microeukaryotic communities, (ii) to explain the community assembly processes and reveal the influencing factors on the balance of the bacterial and microeukaryotic community assembly processes, and (iii) to construct co-occurrence networks of bacterial and microeukaryotic communities and identify the factors affecting them. This study provides a fundamental explanation of microbial community processes and further explains the controlling factors of the evolving underlying mechanism of biodiversity in lake ecosystems on the QTP.

## 2. Materials and Methods

### 2.1. Sample Collection

Briefly, 6 L of water was collected from each site of the 8 sample lakes (at ~ 0.5 m from the top of the water surface) located on the Eastern QTP. Detailed information on the locations is presented in Appendix A and Appendix A. Considering the differences in the catchment areas in each lake, more than two sampling sites were selected at each investigated lake using a portable sterile water sampler in June 2020. The water samples were packed into pre-sterilized glass sampling jars and a plastics bottle for total DNA extraction and the detection of the water’s physicochemical factors’ concentrations. All of the samples were immediately placed in a portable refrigerator at 4 °C after collection. 

### 2.2. Environmental Variable Monitoring 

The water temperature (°C), pH, dissolved oxygen (DO, mg/L), salinity (SAL, g/L), total dissolved solids (TDS, mg/L), electrical conductivity (EC, ms/cm), turbidity (Turb, NTU), oxidation-reduction potential (ORP, mV), and chlorophyll-a (Chl-a, μg/L) were measured in situ three times to generate mean values as the final value of each factor using a portable multiparameter water quality sonde (YSI EXO TM2, 102578, US). In addition, some of the water from each sample was filtered through a 0.45 μm filter, and the filtrates were used to determine the dissolved organic carbon (DOC) concentration using a TOC analysis meter (TOC-LCPH, Shimadzu, Japan) based on the (HJ 501-2009, China). The total nitrogen (TN), ammonia nitrogen (NH_4_^+^-N), nitrate nitrogen (NO_3_^−^-N), nitrite nitrogen (NO_2_^−^-N), and total phosphorous (TP) of each water sample were monitored using gas-phase molecular absorption spectrometry at a detection limit of 0.0003 mg/L according to the standard methods [22]. Meanwhile, the longitude, latitude, and altitude were detected by a portable global positioning system (GPS-China) when the water samples were gathered, and they are listed in Appendix A. 

### 2.3. DNA Extraction, PCR Amplification, and Sequencing Analysis 

The total DNA was extracted according to the methods described in our previous study [22]. The V4 hypervariable region of 16S ribosomal RNA (16S rRNA) genes in the bacteria and the V9 hypervariable region in the microeukaryotes’ 18S ribosomal RNA (18S rRNA) genes were detected using the primers 907R-515F and 1380F-1510R, respectively [16,23]. The composition of PCR amplification and the thermal cycle protocol were performed as described in a previous study, and sequencing was performed on the Illumina Hiseq 2 × 250 platform (Novogene Co., Ltd., Beijing, China) [7]. Briefly, the raw sequencing data of each sample were merged with FLASH (v 1.2.7). The raw reads were filtered and analyzed using QIIME (v 1.9.1) for quality control based on the settings of previous studies [21,24]. The16S rRNA and 18S rRNA trimmed sequences data were grouped into the same operational taxonomic units (OTUs) with UPARSE using a similarity cut off of more than 97% cutoff, and then the taxonomic assignment and alignment were annotated in the SILVA (v 138) and Protist Ribosomal Reference (PR2) databases with a 0.8 confidence threshold using the Ribosomal Databased Project (RDP) classifier (http://rdp.cme.msu.edu/ (accessed on 14 August 2020)) [15]. Finally, the raw sequencing data for 16S rRNA and 18S rRNA were uploaded to the NCBI Sequence Read Archive (SRA) database with the accession numbers PRJNA728144 and PRJNA924075.

### 2.4. Biodiversity Analysis 

To determine the richness and diversity of the bacterial and microeukaryotic communities of the investigated lakes, the alpha diversity indices, including Shannon–Wiener index, Simpson’s diversity index, Chao1 index, Faith’s phylogenetic diversity index, and the Pielou evenness index, were calculated based on the OTUs and phylogeny levels. Spearman’s correlation analyses were conducted to test the relationship between the alpha diversity indices of the microbial community and the environmental factors [24,25,26]. The Kruskal–Wallis test was used to compare the differences in the alpha diversity indices of the microbial community and environmental factors among the 8 lakes [27]. Further, the Mann–Whitney U test was performed to compare the differences in the alpha diversity indices between the bacteria and microeukaryotes. Additionally, we visualized the Bray–Curtis distance matrix to assess the difference in microbial community composition among the different samples using the principal coordinates analysis (PCoA) and the analysis of similarity (ANOSIM) [15]. Redundancy analysis (RDA) was performed to analyze the relationship between the microbial communities and the environmental factors. Variable inflation factors (VIFs) of less than 10 were selected for RDA analysis using the “vegan” packages. To simplify the RDA model, the significant environmental factors, after being checked with the “envfit” functions, were shown in RDA. In addition, variation partition analysis (VPA) was conducted to investigate the pure contributions of spatial factors, water physicochemical factors, and their combined effects on the variations in the bacterial and microeukaryotic community compositions. The spatial factors were generated using the principal coordinates of neighbor matrices (PCNM) analysis in VPA [28]. 

### 2.5. Microbial Community Assembly Processes Analysis 

The neutral model, which estimates the contribution of stochastic processes to microbial assembly, was used to predict the relationship between the frequency of the detected OTU and the mean abundance [29]. The m and R^2^ in the neutral model indicated the estimated migration rate of species and the fitness of the neutral model. Higher m and R^2^ values indicated a lower dispersal limitation of microbial communities and a higher contribution of stochastic processes in microbial assembly processes [30]. Moreover, to further quantify the relative contributions of the stochastic and deterministic processes, the null model based on the phylogenetic β-nearest taxon index (β-NTI) was calculated using the “picante” packages in R [31,32,33]. In the null model, values of β-NTI > 2 or β-NTI < −2 indicated that the deterministic process was the dominant process in regulating microbial community assembly, which can further be divided into homogeneous selection (β-NTI < −2) and heterogeneous selection (β-NTI > 2). When the β-NTI value of −2 < β-NTI < 2, the microbial community assembly is governed by stochastic processes [32]. To further quantify the stochastic process, the value of the Raup–Crick index (RCbray) was enumerated and classified into three ecological processes, including homogeneous dispersal (RCbray < −0.95), dispersal limitation (RCbray > 0.95), and undominated processes (−0.95 < RCbray < 0.95) [34]. Finally, to fully understand the variation in the assemblage processes along environmental gradients, we examined the Spearman correlations between the NTI and the differences in the significant driving factors of the microbial community (Euclidean distance matrices) [19,35]. All of the above analyses were performed using R (v 4.0.5).

### 2.6. Co-Occurrence Network Analysis 

The empirical networks of the bacterial and microeukaryotic communities were generated in “igraph” packages [36]. To simplify the networks, all pairwise Spearman’s rank correlations among the OTUs were calculated. Only robust (r > 0.8 or r < −0.8) and statistically significant correlations (Benjamini–Hochberg-adjusted *p*-values < 0.01) were selected for analysis of the microbial co-occurrence network analysis. Meanwhile, 1000 Erdos–Renyi random networks that had the same number of nodes and edges as the empirical networks were used [37]. The topological properties of the empirical and the random networks of the bacterial and microeukaryotic communities, including the average clustering coefficient, average degree, average path distance, and graph density, were calculated using the “igraph” packages. The associations between the network structures and the measured environmental factors were investigated using the packages of “psych” and “corrplot”. The network visualizations were generated with Gephi (v 0.9.2) software.

## 3. Results

### 3.1. Variations in Relevant Environmental Factors 

The basic values of the water physicochemical factors were measured at each sampled site. All physicochemical factors were measured in triplicate at least. The mean values were the final results, and they are shown in Appendix A. The pH threshold of the lakes varied (8.87–10.10). The water temperature values lay in the range of 1.78–18.17 °C (Appendix A). The mean values of the DOC concentrations were measured, and the highest was found in L3 (Kelvke Lake), with a value of 17.78 mg/L. Also, the contents of nutrients’ concentrations (TN, NH_4_^+^-N, and NO_3_^−^-N) were the highest in L1 (Qinghai Lake). There were significant differences in the water environmental parameter (EC, TDS, SAL, pH, Turb, DOC, TP, TN, NH_4_^+^-N, and NO3^−^-N) concentrations (non-parameters Kruskal–Wallis test, *p* < 0.05, Appendix A) among the eight sampled lakes, whereas there were no significant differences in Chl-a, ORP, and NO_2_^−^-N among the eight investigated lakes.

### 3.2. Microbiological Features 

After the 16S rRNA and 18S rRNA gene amplicon sequencing data trimming, a total number of 8647 and 7474 OTUs of bacterial and microeukaryotic communities were annotated across all samples. The unique and shared OTUs in the bacterial and microeukaryotic communities were different in each investigated lake (Figure 1). The alpha diversity indices of the bacterial and microeukaryotic communities in each lake were computed, tested for significance, and tabulated, as shown in Appendix A. The bacterial communities showed higher species richness (i.e., Chao1) and diversity (i.e., Shannon index) compared to the microeukaryotic communities (Appendix A). Specifically, the highest diversity for bacterial richness (2282) was observed in L1, and the lowest richness was in L8 (1211). The microeukaryotic richness was highest in L5 (2055) and the lowest in L2 (1114). According to the Shannon index, the bacterial communities differentiated among the lakes significantly (*p* = 0.021), whereas the microeukaryotic communities showed insignificant differences among the sampled lakes (Kruskal–Wallis test, *p* = 0.068). Only a significantly higher Chao1 index (*p* < 0.01) was found in the bacterial communities compared to that in microeukaryotic communities (Appendix A). 

The results of the Spearman’s correlations between the water physicochemical factors and the microbial alpha diversity indices revealed that the temperature had a significant positive correlation with the bacterial alpha diversity indices (*p* < 0.001), while it showed insignificant correlations with the microeukaryotic alpha diversity indices (*p* > 0.05, Figure 2). Both the bacterial and microeukaryotic richness and diversity showed the significant relationship with the DOC and TN (*p* < 0.01).

The predominant prokaryotic phylum in each investigated lake was Proteobacteria, with a proportion of 41.78%, and the most abundant classes of microeukaryotic community in the distinct sampled sites differed (Appendix A). For example, Chrysophyceae (15.11%) was dominated in QHL1. Considering the beta-diversity of the bacterial and microeukaryotic communities, PCoA based on the Bray–Curtis distance for bacterial and microeukaryotic communities was performed, and the results showed that the most important eigenvalues of the first two axes explained 44.58% and 38.15% of the bacterial and microeukaryotic community variations, respectively, and the compositions in both the bacterial and microeukaryotic communities were diverse among the sampled lakes (Appendix A). ANOSIM analysis revealed that the bacterial and microeukaryotic community compositions were significantly different among some sampled lakes (Appendix A). For example, the bacterial and microeukaryotic community compositions in Qinghai Lake showed remarkable differences from the other lakes (*p* < 0.05).

The relationships between the microbial compositions and the environmental factors were evaluated by using RDA, and overall variabilities of 58.68% and 32.57% of the bacterial and microeukaryotic community compositions were found, respectively (Figure 2). Clearly, the latitude and pH were the most significant environmental factors in shaping the bacterial and microeukaryotic community compositions, respectively. Based on the RDA results, we found that the variance of the bacterial community composition was mainly driven by altitude and latitude (Figure 3). For microeukaryotic community composition, the SAL was the most important environmental factor in the positive direction of the first axis (Figure 3).

The VPA results indicate that the pure contributions of spatial factors accounted for a higher part of the variability in microeukaryotic community composition (11%), which indicates that the microeukaryotic communities suffered stronger dispersal limitation (represented by spatial factors) compared to the bacterial communities (3%). The variance explained uniquely by the water physicochemical factors in the bacterial community composition and microeukaryotic community composition reached 19% and 14%, respectively (Figure 4). Moreover, the combined variation in both the spatial factors and water physicochemical factors was lower in the microeukaryotic community composition than that in the bacterial community composition. The fractions that were left unexplained were 60% for the bacterial communities and 61% for the microeukaryotic communities.

### 3.3. Disentangling the Microbial Assembly Processes

As indicated by Figure 5, the neutral model estimated the moderate relationships between the occurrence frequency of the bacterial and microeukaryotic OTUs and their mean relative abundance. The goodness of the neutral model varied in the bacterial (R^2^ = 0.673) and microeukaryotic communities (R^2^ = 0.628), respectively. The m value was calculated to be 0.089 in the bacterial community and 0.083 in the microeukaryotic community, suggesting that the bacterial species’ dispersal ability was higher than that of the microeukaryotic species.

Furthermore, the null model indicates that the bacterial community was largely regulated by homogeneous selection (56.40%), followed by dispersal limitation processes (23.65%), homogeneous dispersal (1.72%), heterogeneous selection (0.50%), and undominated (17.71%, Figure 6). These suggest that the bacterial community composition tended to be similar under the influence of a homogeneous selection process. In contrast, the dispersal limitation was (67.24%), the most dominant process in assembling the microeukaryotic community (Figure 6).

In order to further infer the changes in the assemblage deterministic and stochastic processes, we examined the relationship between the Beta-NTI and the significant water physicochemical factors in the microbial variations (pH, DO, DOC, Temp, NH_4_^+^-N, and SAL). Overall, the pH was significantly affiliated with the Beta-NTI in both the bacterial and microeukaryotic community structures’ assemblies. For the bacterial community, the Beta-NTI showed a significantly positively relationship with the DOC, indicating increasing differences in the DOC, resulting in a shift from homogeneous selection (Beta-NTI < −2) to stochastic assembly processes (Figure 7). For the microeukaryotic community, the SAL was more strongly related to the Beta-NTI than other factors (Figure 8).

### 3.4. Network Analysis of Microbial Communities

On the whole, the empirical networks of bacterial and microeukaryotic communities revealed that the number of positive correlations in the networks was much higher than negative ones, suggesting that both the bacterial and microeukaryotic communities were formed mainly by niche sharing (Figure 9, Table 1). Both the empirical bacterial and microeukaryotic communities and their random networks exhibited a modular structure (modularity > 0.4), while the empirical networks’ topological parameters (i.e., modularity, average degree, and average path length) in both the bacterial and microeukaryotic communities had higher values than those in their random networks. Thus, the bacterial and microeukaryotic community networks were constructed non-randomly (Table 1). Moreover, we assessed the relationship between the networks’ properties and environmental factors. Specifically, there was a significant trend of decreasing network complexity properties with increasing EC, TDS, SAL, DOC, TN, and NO_3_^−^-N in both the bacterial and microeukaryotic communities (Figure 10). Additionally, the average path length of the bacterial community network increased significantly (*p* < 0.001) with NH_4_^+^-N and decreased significantly with altitude (*p* < 0.001, Figure 10).

## 4. Discussion

We analyzed the diversity and community compositions, drivers, ecological assembly processes, and co-occurrence networks of bacterial and microeukaryotic communities from 29 sampled sites in 8 investigated lakes on the QTP. 

A significantly higher Chao1 index (*p* < 0.01) was found in the bacterial community than in the microeukaryotic community, showing a significantly positive relationship with the DOC, TN, and NO_3_^−^-N, whereas the Chao1 index of the microeukaryotic community had negative correlations with the same factors (Figure 2). The possible explanation for this is that the common limiting macronutrients in waters could limit the primary productivity of microeukaryotic communities (i.e., algae). Accordingly, the competitions among microeukaryotic species for nutrients can lead to a decrease in their richness [38]. 

Furthermore, the variations in the bacterial and microeukaryotic communities significantly responded to the distinct environmental factors (i.e., altitude and SAL), with the exceptions of latitude and pH (Figure 3). Congruent with our results, many prior studies have shown that the pH, DOC, and NH_4_^+^-N are significant drivers of bacterial communities in lake ecosystems on the QTP [35,39,40]. According to a previous study, the pH influenced bacterial community composition by changing the nutrient availability or organic carbon characteristics [11]. However, Zhong et al. [40] and Wu et al. [41] revealed no significant effects of altitude on bacterial community structure in the plateau lakes of the Tibetan Plateau, and this is inconsistent with our findings. This might be because all investigated lakes in previous studies were located at similar altitudes. For the microeukaryotic community, it has been demonstrated that changes in the pH are expected to lead to shifts in calcifying algal community composition, and the SAL has a positive relationship with microeukaryotes’ extracellular osmolarity, meaning that microeukaryotic communities that are not able to adapt to osmotic stress will possibly die or become less active [16,38].

The null model results suggested that homogenous selection plays a more important role in determining the composition of bacterial communities, while the dispersal limitation overwhelms deterministic processes (homogeneous selection and homogeneous dispersal) in structuring microeukaryotic community composition (Figure 6). These findings are in agreement with previous studies [14,21,42]. This implies that niche-related processes play a more influential role than neutral processes in regulating the assembly of bacterial communities [14], and the bacterial community compositions in these investigated lakes tend to be similar [12]. 

Furthermore, the estimated immigration rate (m) of the bacterial communities was slightly higher than that of the microeukaryotic communities, implying the more serious dispersal limits suffered by the microeukaryotic communities. These results are also supported by the “size-dispersal” hypothesis, which states that compared to a larger body size of microeukaryotes, smaller bacteria may be less susceptible to the damage caused by the turbulences and debris during migration, and, therefore, have a higher migration rate [15,21,43]. 

In low-altitude aquatic ecosystems, previous studies have reported that homogeneous selection and heterogeneous selection are predominant in shaping bacterial communities [12,14,42]. However, we found that bacterial community assembly in the high-altitude lakes was mainly governed by homogeneous selection (Figure 6). This shift in ecological selection may be attributed to the regional differences in deterministic factors (i.e., nutrients concentrations, altitude, and water temperature). Furthermore, variable selection accounts for a larger part of the bacterial community assembly processes in Tibetan lakes according to prior studies [18,21], while our findings reveal that variable selection only explains approximately 0.5% of the bacterial assembly processes. The possible explanation for this difference is that the geographic scales and environmental gradients largely determine the balance between deterministic and stochastic processes [28]. For example, Zeng et al. [42] reported that the relative importance of stochastic processes governing the assembly of the bacterial community in freshwater lakes enhanced with eutrophication increased. Additionally, our results indicate significant correlations between the Beta-NTI metric degree of bacterial and microeukaryotic communities (considered as community turnover) and several in situ water physicochemical factors, indicating that small variations in habitat conditions can exert some measurable changes on microbial community composition, which is in agreement with the bacterial and archaeal community assembly processes in Ohio aquifers [44]. Specifically, the DOC and SAL were the most influential factors for determining the bacterial and microeukaryotic community assembly processes, respectively (Figure 7 and Figure 8). This implies that the DOC and SAL regulated the balance of deterministic and stochastic assembly processes for bacterial and microeukaryotic communities, respectively, which is in agreement with a previous study on a subtropical urban reservoir [16].

To further identify the interspecies correlations of microbial communities, the empirical and their random networks in the bacterial and microeukaryotic communities were established (Figure 9, Table 1). The proportions of positive relationships among the bacterial community empirical networks (92.94%) was lower than that in the microeukaryotic community empirical networks (Table 1), which implies that the bacterial species had lower rates of cooperation, parasitism, and symbiosis [21]. This may be caused by a stronger competition among the microeukaryotic taxa for the niche occupation and resources because most microeukaryotes are heterotrophic [38,45]. Additionally, the topological properties in the empirical networks of both the bacterial and microeukaryotic communities were much higher than those in their random networks, indicating that the network structures had non-random features, which is consistent with a previous study [14]. The more complex network in the bacterial taxa (represented by a higher average degree, shorter HD, and higher modularity) suggests that the bacterial network is more interconnected than the microeukaryotic network (Table 1). 

Meanwhile, compared to the microeukaryotic diversity, a higher diversity was found in the bacterial community (Appendix A), and these results are supported by the diversity begets diversity (DBD) model, which predicted a positive effect of diversity on diversification [11,46]. Thus, this finding indicates that microbial interactions could act as a kind of deterministic selection force for microbial community assembly. It also explains the results of the higher contributions of deterministic processes in the bacterial community assembly processes.

## 5. Conclusions

In this study, we investigated the bacterial and microeukaryotic community compositions and their environmental drivers, quantified their community assembly processes, and constructed their co-occurrence networks in lake ecosystems on the QTP. These findings revealed that the Chao1 index in the bacterial community was significantly higher than that in the microeukaryotic community. The altitude and pH were the most significant factors in shaping the bacterial and microeukaryotic community compositions. Meanwhile, homogeneous selection strongly governed the bacterial community compositions, whereas dispersal limitation dominated the processes of the microeukaryotic communities. Importantly, our results reveal the underlying mechanisms mediating the balance between the deterministic and stochastic processes of bacterial and microeukaryotic community compositions. The co-occurrence networks revealed that the bacterial and microeukaryotic communities showed non-random features, and some environmental factors, such as altitude, SAL, and NH_4_^+^-N, influenced the complexity of the networks significantly. Our findings expand the understanding of the underlying mechanisms of microbial community assembly in the plateau lakes.

## Figures and Tables

**Figure 1 microorganisms-12-00032-f001:**
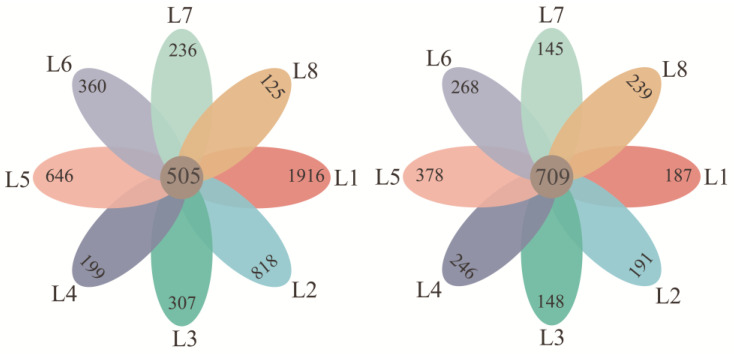
The unique and shared OTUs at a dissimilarity level of less than 3% of bacteria (**left** panel) and microeukaryotes (**right** panel) in each sampled lake. Notations: L1 to L8 represent Qinghai Lake, Tuosu Lake, Kelvke Lake, Hala Lake, Eling Lake, Zhaling Lake, Donggi Cona Lake, and Kusai Lake, respectively.

**Figure 2 microorganisms-12-00032-f002:**
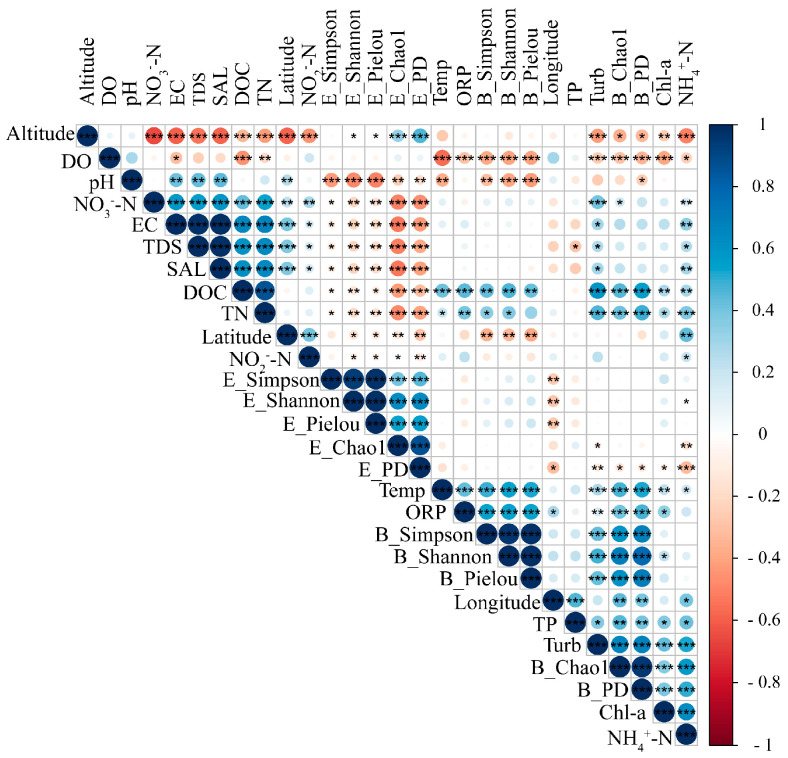
Spearman’s rank correlations among water physicochemical factors and alpha diversity indices of bacterial (B) and microeukaryotic (E) communities. Asterisk symbols indicate the significant relationships: * *p* < 0.05; ** *p* < 0.01; *** *p* < 0.001. Notations: Temp, TDS, EC, ORP, Turb, DO, DOC, SAL, Chl-a, TP, TN, NH_4_^+^-N, NO_3_^−^-N, NO_2_^−^-N, and PD represent temperature, total dissolved solids, electrical conductivity, oxidation–reduction potential, dissolved oxygen, dissolved organic carbon, salinity, chlorophyll-a, total phosphorus, total nitrogen, ammonia nitrogen, nitrate nitrogen, nitrite nitrogen, and Faith’s phylogenetic diversity, respectively.

**Figure 3 microorganisms-12-00032-f003:**
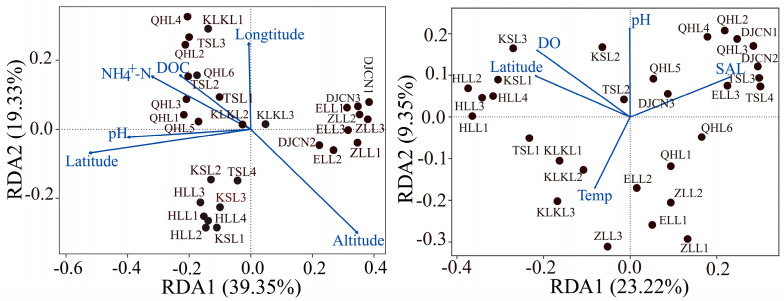
Influence of environmental variables on bacterial (**left** panel) and microeukaryotic (**right** panel) community compositions, as shown by using redundancy analysis. Redundancy analysis was conducted to visualize significant environmental factors affecting bacterial and microeukaryotic community compositions at the OTU level.

**Figure 4 microorganisms-12-00032-f004:**
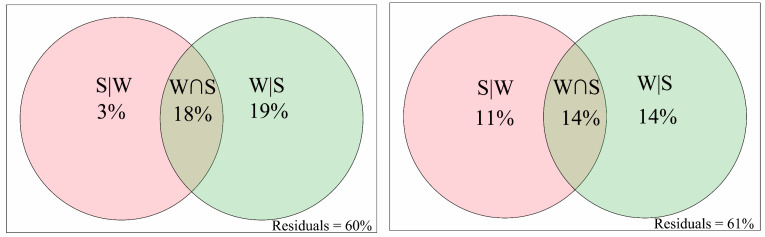
Variation partitioning analysis (VPA) was performed to quantify the pure contribution of PCNM-based spatial (S) and water physicochemical (W) factors to bacterial (**left** panel) and microeukaryotic (**right** panel) community changes.

**Figure 5 microorganisms-12-00032-f005:**
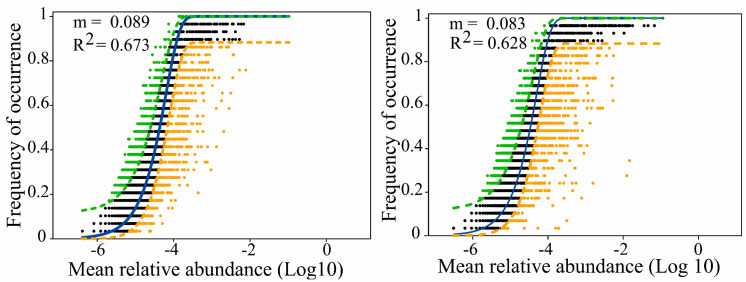
Fitting of the bacterial (**left** panel) and microeukaryotic (**right** panel) community assembly processes using the neutral model. Green and orange dashed lines represent the upper and lower of 95% confidence intervals in the model prediction. Adjusted R^2^ and m indicate the fitness of the neutral model and the estimated migrated rate of the microbial communities, respectively.

**Figure 6 microorganisms-12-00032-f006:**
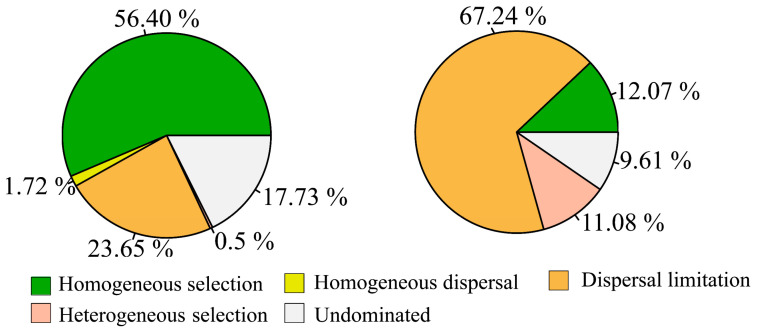
The contributions of deterministic (homogeneous selection and heterogeneous selection) and stochastic processes (dispersal limitation, homogeneous dispersal, and undominated) in bacterial (**left** panel) and microeukaryotic (**right** panel) community assembly processes.

**Figure 7 microorganisms-12-00032-f007:**
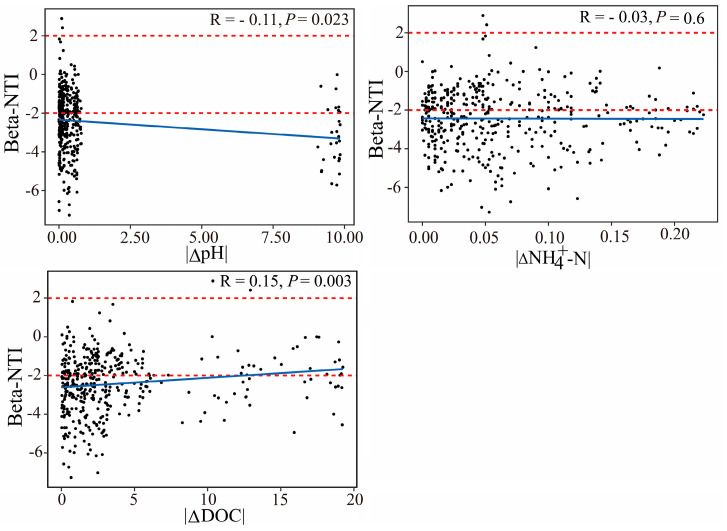
The relationship (Spearman’s) between bacterial community Beta-NTI and differences (Δ) in pH, NH_4_^+^-N, and DOC. Red horizontal dashed lines indicate the Beta-NTI significance thresholds of |2|.

**Figure 8 microorganisms-12-00032-f008:**
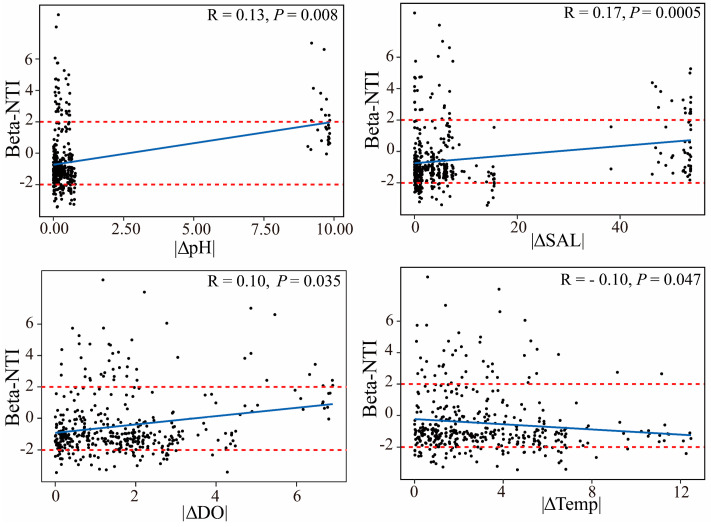
The relationship (Spearman’s) between microeukaryotic community Beta-NTI and differences (Δ) in SAL, Temp, pH, and DO. Red horizontal dashed lines indicate the Beta-NTI significance thresholds of |2|.

**Figure 9 microorganisms-12-00032-f009:**
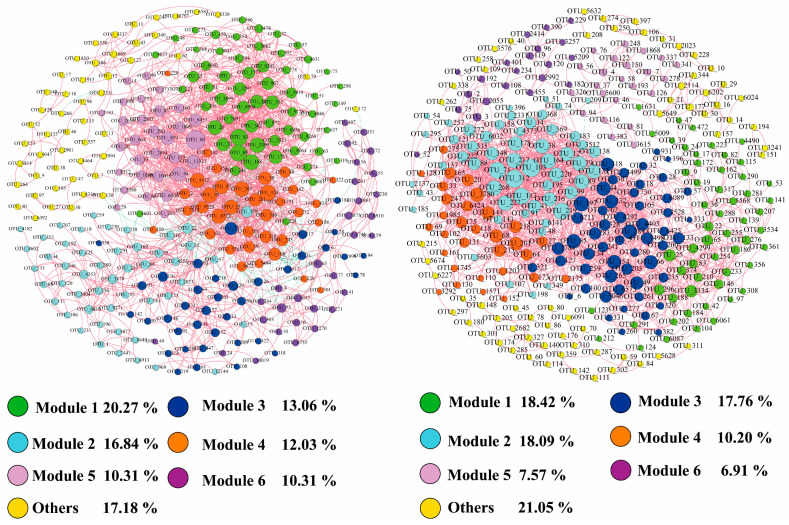
Co-occurrence network analysis of bacterial (**left** panel) and microeukaryotic (**right** panel) communities. The color and size of each node in the networks represent different modules and the degrees of connection based on the OTU level. Red and green edges indicate positive and negative relationships, respectively, between any two OTUs.

**Figure 10 microorganisms-12-00032-f010:**
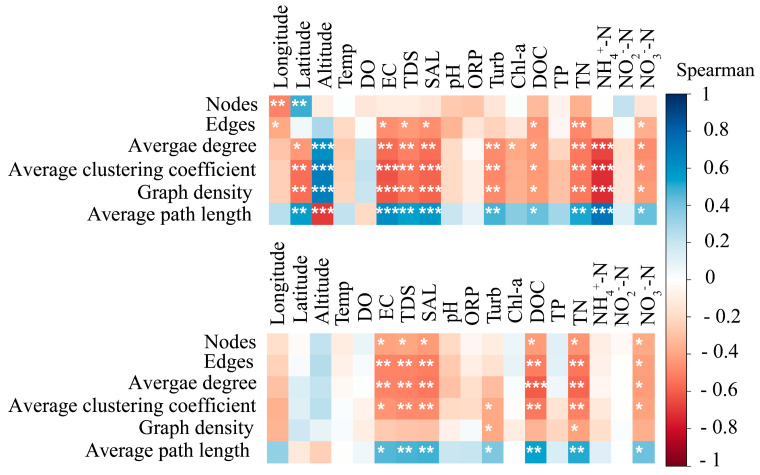
The relationships between the empirical bacterial (top panel) and microeukaryotic (below panel) community networks’ properties and environmental factors. Asterisk symbols indicate the significant relationships: * *p* < 0.05; ** *p* < 0.01; *** *p* < 0.001.

**Table 1 microorganisms-12-00032-t001:** Topological properties of the empirical bacterial and microeukaryotic community networks and their random networks.

	Network Properties	Bacteria	Microeukaryotes
Random network	Average degree (avg K)	5.064	5.382
	Average clustering coefficient (avg CC)	0.05	0.07
	Average path distance (GD)	3.767	3.824
	Geodesic efficiency (E)	0.314	0.311
	Harmonic geodesic distance (HD)	3.181	3.217
	Modularity	0.413	0.459
Empirical network	Nodes	291	304
	Edges	1728	1726
	(Positive/negative%)	(92.94%/7.06%)	(99.97%/0.23%)
	Average degree	11.876	11.335
	Graph density	0.041	0.037
	Modularity	0.627	0.523
	Average path length	3.751	4.873
	Average clustering coefficient	0.538	0.572

## Data Availability

The data presented in this study are available upon request from the corresponding author.

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
