# Peer review of "Bacterial and Microeukaryotic Community Compositions and Their Assembly Processes in Lakes on the Eastern Qinghai-Tibet Plateau"

_microorganisms, 2023, doi:10.3390/microorganisms12010032_

Round 1

Reviewer 1 Report

Comments and Suggestions for Authors

Line 40: The sentence "...QTB, distribution immense...." has to be edited to get complete. For example, "...QTB with a great number...."

Lines 48-51: sentence needs to be edited, a verb is missing

Line 52: "to analyse" instead of "to analysis"

line 60: "were affected" instead of "were affect"

Line 68: ?mounting studies

''''

Line 256: "the most factors in the positive direction of the first axis was?"

Comments on the Quality of English Language

Very bad, unreadable English

Reviewer 2 Report

Comments and Suggestions for Authors

The manuscript “Bacterial and Microeukaryotic Community Composition and their Assembly Processes in Lakes on the Eastern of the Qinghai-Tibet Plateau” by Wang et al. presents a good overview of the diversity and microbial community compositions of 8 lakes in the lake ecosystems on the Qinghai-Tibet Plateau (QTP). The authors also investigated the driving factors, ecological assembly processes, and networks of co-occurrence of prokaryotes and eukaryotes in the water bodies. The authors' observations provide some deeper insights into the understanding of microbial community composition and biodiversity. 

The presented results are of broad interest, relatively concise and novel and, therefore, I would recommend the manuscript to be accepted for publication in “Microorganisms” after some modifications: 

It would be nice to see the depth of the sequencing approach. Please provide the information on raw  and processed data (maybe rarefaction curves) in the supplemental material?

Line 425:  “…most of microeukaryotes are heterotrophic” based on what results? According to Fig S4 (right panel) only a few samples were predominated by heterotrophic microeukaryotes, and a large proportion of samples were phytoplanktonic (photoautotrophs). Besides a large proportion of the communities (~50%) is marked as “others”. Please explain!

In general, it could be a valuable information to get some deeper insight into the composition of the communities not just on phyla (for bacteria) or class (for microeukaryotes) levels. And maybe compare the most abundant taxa? Would be there any correlations?

Figure 8 in the manuscript should be presented much larger since it is very difficult to distinguish the different colors.

Comments on the Quality of English Language

The pdf file seems to have some formatting problems, since a lot of words have been transferred unproperly to the next line. The manuscript contains also many minor grammatical and wording mistakes that need to be corrected, e.g. line 24 (…were differed …), line 60 (…were affect…) and so on. Please go very carefully through the manuscript and correct accordingly!

Line 43: … intense radiation and oligotrophy (not oligotrophic). Microbials (or microorganisms or microbial communities) harbored…  Please change accordingly.

Line 52: maybe better: “To date, numerous studies have used co-occurrence network to analyse species-species interactions of in the lake ecosystems.”

Lines 74-76: Please rephrase the sentence.

Lines 79-81: maybe better: “However, fewer studies to date have focused on the influences of shifting environmental factors on the balance of deterministic and stochastic processes of microbial community formation in lake ecosystems on the QTP.”

Line 87: It should be either: This study… or The present study… will provide a fundamental comprehensive what? in structuring microbial community processes…? Please rephrase!

Line 107-108: it should be 0.45µm?

Line 113: Please cite the literature for the standard methods used for the measurements of N (and subspecies) and TP.

Line 445: Meanwhile, Homologeneous selection… Please correct!

Reviewer 3 Report

Comments and Suggestions for Authors

The article is devoted to attempts to understand the mechanism of development of microbial communities in the water of lakes located on the Qinghai Tibetan Plateau. The authors obtained and analyzed extensive material, but questions arose that need clarification.

1. What explains the choice of this particular volume of water for DNA analysis? Why is it the same for all lakes? Although water characteristics vary significantly in lakes. How was the sterility of sampling ensured?

2. It is necessary to clarify what the authors mean by microeukaryotes. How did the authors manage to separate the DNA of microeukaryotes and just eukaryotes using the primers they used?

3. The Figure S2 from the supplementary materials should be transferred to the main text, and the supplementary materials should contain a list of all prokaryotes and microeukaryotes at the phylum level found in lake water samples, and not just the top 10 at the phylum level, maybe in the form of an Excel file. So far, judging by the figure, proteobacteria were not the most represented everywhere. In some samples, other phyla predominated. These results should be presented in detail in the text, and not as schematically as the authors did.

4. All Latin names of microorganisms, even high-ranking taxa, must be written in italics.

5. The entire text is printed with breaks in the last words in the line, which makes it very difficult to read and perceive.

Round 2

Reviewer 3 Report

Comments and Suggestions for Authors

I like this version of the manuscript much better. However, there are still word breaks everywhere at the end of each line. This technical problem needs to be resolved.

Author Response

Dear reviewer:
